# ExpandNets: Linear Over-parameterization to Train Compact Convolutional Networks

**Shuxuan Guo**
CVLab, EPFL
Lausanne 1015, Switzerland
shuxuan.guo@epfl.ch

**Jose M. Alvarez**
NVIDIA
Santa Clara, CA 95051, USA
josea@nvidia.com

**Mathieu Salzmann**
CVLab, EPFL
Lausanne 1015, Switzerland
mathieu.salzmann@epfl.ch

## Abstract

We introduce an approach to training a given compact network. To this end, we leverage over-parameterization, which typically improves both neural network optimization and generalization. Specifically, we propose to expand each linear layer of the compact network into multiple consecutive linear layers, without adding any nonlinearity. As such, the resulting expanded network, or ExpandNet, can be contracted back to the compact one algebraically at inference. In particular, we introduce two convolutional expansion strategies and demonstrate their benefits on several tasks, including image classification, object detection, and semantic segmentation. As evidenced by our experiments, our approach outperforms both training the compact network from scratch and performing knowledge distillation from a teacher. Furthermore, our linear over-parameterization empirically reduces gradient confusion during training and improves the network generalization.

## 1 Introduction

With the growing availability of large-scale datasets and advanced computational resources, convolutional neural networks have achieved tremendous success in a variety of tasks, such as image classification [17, 28], object detection [39, 40, 42] and semantic segmentation [35, 45]. Over the past few years, "Wider and deeper are better" has become the rule of thumb to design network architectures [17, 22, 51, 52]. This trend, however, raises memory- and computation-related challenges, especially in the context of constrained environments, such as embedded platforms.

Deep and wide networks are well-known to be heavily over-parameterized, and thus a compact network, both shallow and thin, should often be sufficient. Unfortunately, compact networks are notoriously hard to train from scratch. As a consequence, designing strategies to train a given compact network has drawn growing attention, the most popular approach consisting of transferring the knowledge of a deep teacher network to the compact one of interest [19, 18, 20, 37, 44, 54, 60, 61].

In this paper, we introduce an alternative approach to training compact neural networks, complementary to knowledge transfer. To this end, building upon the observation that network over-parameterization improves both optimization and generalization [1, 2, 5, 26, 41, 49, 62], we propose to increase the number of parameters of a given compact network by incorporating additional layers. However, instead of separating every two layers with a nonlinearity, we advocate introducing consecutive *linear* layers. In other words, we expand each linear layer of a compact network into a succession of multiple linear layers, without any nonlinearity in between. Since consecutive linear layers are equivalent to a single one [50], such an expanded network, or ExpandNet, can be algebraically contracted back to the original compact one without any information loss.

While the use of successive linear layers appears in the literature, existing work [5, 6, 25, 29, 50, 63] has been mostly confined to *fully-connected networks without any nonlinearities* and to the theoretical study of their behavior under fairly unrealistic statistical assumptions. In particular, these studies aim

to understand the learning dynamics and the loss landscapes of deep networks. Here, by contrast, we focus on *practical, nonlinear, compact convolutional* neural networks, and demonstrate the use of linear expansion as a means to introduce over-parameterization and facilitate training, so that a given compact network achieves better performance.

Specifically, as illustrated by Figure 1, we introduce three ways to expand a compact network: (i) replacing a $k \times k$ convolution by three convolutional layers with kernel size $1 \times 1$, $k \times k$ and $1 \times 1$, respectively; (ii) replacing a $k \times k$ convolution with $k > 3$ by multiple $3 \times 3$ convolutions; and (iii) replacing a fully-connected layer with multiple ones. Our experiments demonstrate that expanding convolutions is the key to obtaining more effective compact networks.

In short, our contributions are (i) a novel approach to training a given compact, nonlinear convolutional network by expanding its linear layers; (ii) a strategy to expand convolutional layers with arbitrary kernels; and (iii) a strategy to expand convolutional layers with kernel size larger than 3. We demonstrate the benefits of our approach on several tasks, including image classification on ImageNet, object detection on PASCAL VOC and image segmentation on Cityscapes. Our ExpandNets outperform both training the corresponding compact networks from scratch and using knowledge distillation. We empirically show over-parameterization to be the key factor for such better performance. Furthermore, we analyze the benefits of linear over-parameterization during training via experiments studying generalization, gradient confusion and the loss landscape. Our code is available at `https://github.com/GUOShuxuan/expandnets`.

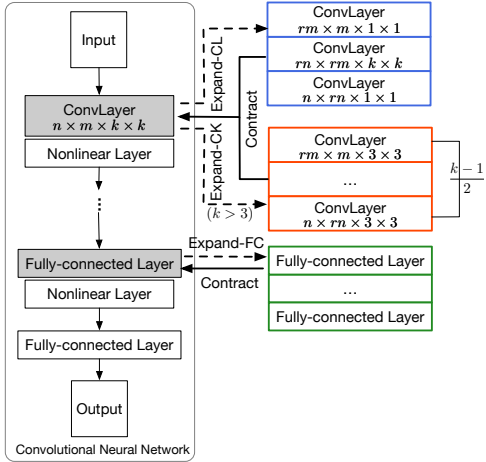

Figure 1: **ExpandNets.** We propose 3 strategies to linearly expand a compact network. An expanded network can then be contracted back to the compact one algebraically, and outperforms training the compact one, even with knowledge distillation.

## 2  Related Work

Very deep convolutional neural networks currently constitute the state of the art for many tasks. These networks, however, are known to be heavily over-parameterized, and making them smaller would facilitate their use in resource-constrained environments, such as embedded platforms. As a consequence, much research has recently been devoted to developing more compact architectures.

Network compression constitutes one of the most popular trends in this area. In essence, it aims to reduce the size of a large network while losing as little accuracy as possible, or even none at all. In this context, existing approaches can be roughly grouped into two categories: (i) parameter pruning and sharing [7, 9, 14, 16, 31, 32, 38, 55], which aims to remove the least informative parameters, yielding an arbitrary compact network with information loss; and (ii) low-rank matrix factorization [10, 24, 30, 34, 47], which uses decomposition techniques to reduce the size of the parameter matrix/tensor in each layer. While compression is typically performed as a post-processing step, it has been shown that incorporating it during training could be beneficial [3, 4, 56, 57]. In any event, even though compression reduces a network's size, it neither provides one with the flexibility of designing a network with a specific architecture, nor incorporates over-parameterization to improve the performance of compact network training. Furthermore, it often produces networks that are much larger than the ones we consider here, e.g., compressed networks with $O(1M)$ parameters vs $O(10K)$ for the SmallNets used in our experiments.

In a parallel line of research, several works have proposed design strategies to reduce a network's number of parameters [21, 36, 43, 48, 53, 58]. Again, while more compact networks can indeed be developed with these mechanisms, they impose constraints on the network architecture, and thus do not allow one to simply train a given compact network. Furthermore, as shown by our experiments, our approach is complementary to these works. For example, we can improve the results of MobileNets [21, 48] by training them using our expansion strategy.

Here, in contrast to the above-mentioned literature, we seek to train a *given* compact network with an arbitrary architecture. This is also the task addressed by knowledge transfer approaches [18–20, 37, 44, 54, 60, 61]. To achieve this, existing methods leverage the availability of a pre-trained very deep teacher network. In this paper, we introduce an alternative strategy to train compact networks, complementary to knowledge transfer. Inspired by the theory showing that over-parameterization helps training [1, 2, 5, 26, 41, 49, 62], we expand each linear layer in a given compact network into a succession of multiple linear layers. Our experiments evidence that training such expanded networks, which can then be contracted back algebraically, yields better results than training the original compact networks, thus empirically confirming the benefits of over-parameterization. Our results also show that our approach outperforms knowledge distillation, even without using a teacher network.

Note that linearly over-parameterized neural networks have been investigated both in the early neural network days [6] and more recently [5, 15, 25, 29, 50, 63]. These methods, however, typically study purely linear networks, with a focus on the convergence behavior of training in this linear regime. For example, Gunasekar et al. [15] demonstrated that a different parameterization of the same model dramatically affects the training behavior; Arora et al. [5] showed that linear over-parameterization modifies the gradient updates in a unique way that speeds up convergence; Wu et al. [59] collapsed multiple FC layers into a single one by removing the non-linearities from the *MLP* layers of a graph convolutional network. In contrast to these methods, which focus on fully-connected layers, we develop two strategies to expand *convolutional* layers, and empirically demonstrate the impact of our expansion strategies on prediction accuracy, training behavior and generalization ability.

The concurrent work ACNet of Ding et al. [11] also advocates for expansion of convolutional layers. However, the two strategies we introduce differ from their use of 1D asymmetric convolutions, and our experiments show that our approach outperforms theirs. More importantly, this work constitutes further evidence of the benefits of linear expansion.

## 3 ExpandNets

Let us now introduce our approach to training compact networks by linearly expanding their layers. Below, we focus on our two strategies to expand convolutional layers, and then briefly discuss the case of fully-connected ones.

### 3.1 Expanding Convolutional Layers

We propose to linearly expand a convolutional layer by replacing it with a series of convolutional layers. To explain this, we will rely on the fact that a convolution operation can be expressed in matrix form. Specifically, let $\mathbf{X}_{b \times m \times w \times h}$ be the input tensor to a convolutional layer, with batch size $b$, $m$ input channels, height $h$ and width $w$, and $\mathbf{F}_{n \times m \times k \times k}$ be the tensor encoding the convolutional filters, with $n$ output channels and kernel size $k$. Ignoring the bias, which can be taken into account by incorporating an additional channel with value 1 to $\mathbf{X}$, a convolution can be expressed as

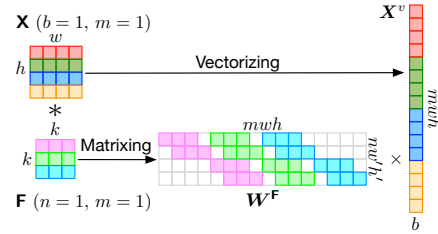

Figure 2: **Matrix representation of a convolutional layer** (best viewed in color).

$$\mathbf{Y}_{b \times n \times w' \times h'} = \mathbf{X}_{b \times m \times w \times h} * \mathbf{F}_{n \times m \times k \times k} = \text{reshape}(\boldsymbol{W}^{\mathbf{F}}_{nw'h' \times mwh} \times \boldsymbol{X}^v_{mwh \times b}), \qquad (1)$$

where $\mathbf{Y}_{b \times n \times w' \times h'}$ is the output tensor, $X^v_{mwh \times b}$ is a matrix representation of $\mathbf{X}$, and $W^{\mathbf{F}}_{nw'h' \times mwh}$ is a highly structured sparse matrix containing the convolutional filters. This process, which is a bijection, is depicted by Figure 2 for $b = 1$, $m = 1$ and $n = 1$.

With this matrix representation, one can therefore expand a layer linearly by replacing the matrix $\boldsymbol{W}^{\mathbf{F}}$ with a product of an arbitrary number of matrices. However, using arbitrary matrices would ignore the convolution structure, and thus alter the original operation performed by the layer. Fortunately, multiplying several convolution matrices still yields a valid convolution, as can be confirmed by observing the pattern within the matrix in Figure 2. Nevertheless, one cannot simply expand a convolutional layer with kernel size $k \times k$ as a series of convolutions with arbitrary kernel sizes because, in general, the resulting receptive field size would differ from the original one. To overcome this, we propose the two expansion strategies discussed below.

**Expanding general convolutions.** For our first strategy, we note that $1 \times 1$ convolutions retain the computational benefits of convolutional layers while not modifying the receptive field size. As illustrated in Figure 1, we therefore propose to expand a $k \times k$ convolutional layer into 3 consecutive convolutional layers: a $1 \times 1$ convolution; a $k \times k$ one; and another $1 \times 1$ one. Importantly, this allows us to increase not only the number of layers, but also the number of channels by setting $p, q > n, m$. To this end, we rely on the notion of *expansion rate*. Specifically, for an original layer with $m$ input channels and $n$ output ones, given an expansion rate $r$, we define the number of output channels of the first $1 \times 1$ layer as $p = rm$ and the number of output channels of the intermediate $k \times k$ layer as $q = rn$. Note that other strategies are possible, e.g., $p = r^i m$, but ours has the advantage of preventing the number of parameters from exploding.

Once such an expanded convolutional layer has been trained, one can contract it back to the original one algebraically by considering the matrix form of Eq. 1. That is, given the filter tensors of the intermediate layers, $\mathbf{F}^1_{p \times m \times 1 \times 1}$, $\mathbf{F}^2_{q \times p \times k \times k}$ and $\mathbf{F}^3_{n \times q \times 1 \times 1}$, the matrix representation of the original layer can be recovered as

$$\boldsymbol{W}^{\mathbf{F}}_{nw'h' \times mwh} = \boldsymbol{W}^{\mathbf{F}^3}_{nw'h' \times qw'h'} \times \boldsymbol{W}^{\mathbf{F}^2}_{qw'h' \times pwh} \times \boldsymbol{W}^{\mathbf{F}^1}_{pwh \times mwh} \,, \tag{2}$$

which encodes a convolution tensor. At test time, we can thus use the original compact network. Because it applies to any size $k$, we will refer to this strategy as *expanding convolutional layers*.

**Expanding $k \times k$ convolutions with $k > 3$.** While $3 \times 3$ kernels are popular in very deep architectures [17], larger kernel sizes are often exploited in compact networks, so as to increase their expressiveness and their receptive fields. As illustrated in Figure 1, $k \times k$ kernels with $k > 3$ can be equivalently represented with a series of $l$ $3 \times 3$ convolutions, where $l = (k-1)/2$. Note that $k$ is typically odd in CNNs. We then have

$$\mathbf{Y} = \mathbf{X} * \mathbf{F}_{n \times m \times k \times k} = \mathbf{X} * \mathbf{F}^1_{p_1 \times m \times 3 \times 3} * \cdots * \mathbf{F}^{l-1}_{p_{l-1} \times p_{l-2} \times 3 \times 3} * \mathbf{F}^l_{n \times p_{l-1} \times 3 \times 3} \,. \tag{3}$$

As before, the number of channels in the intermediate layers can be larger than that in the original $k \times k$ one, thus allowing us to linearly over-parameterize the model. For an expansion rate $r$, we set the number of output channels of the first $3 \times 3$ layer to $p_1 = rm$ and that of the subsequent layers to $p_i = rn$. The same matrix-based strategy as before can be used to algebraically contract back the expanded unit into $\mathbf{F}_{n \times m \times k \times k}$. We will refer to this strategy as *expanding convolutional kernels*.

## 3.2 Expanding Convolutions in Practice

**Padding and strides.** In modern convolutional networks, padding and strides are widely used to retain information from the input feature map while controlling the size of the output one. To expand a convolutional layer with padding $p$, we propose to use padding $p$ in the first layer of the expanded unit while not padding the remaining layers. Furthermore, to handle a stride $s$, when expanding convolutional layers, we set the stride of the middle layer to $s$ and that of the others to 1. When expanding convolutional kernels, we use a stride 1 for all layers except for the last one whose stride is set to $s$. These two strategies guarantee that the resulting ExpandNet can be contracted back to the compact model without any information loss.

**Depthwise convolutions.** Depthwise convolutions are often used to design compact networks, such as MobileNet [21], MobileNetV2 [48] and ShuffleNetV2 [36]. To handle them, we make use of our general convolutional expansion strategy within each group. Specifically, we duplicate the input channels $r$ times and employ cross-channel convolutions within each group. This makes the expanded layers equivalent to the original ones.

## 3.3 Expanding Fully-connected Layers

Beacuse the weights of a fully-connected layer can naturally be represented in matrix form, our approach directly extends to such layers. That is, we can expand a fully-connected layer with $m$ input and $n$ output dimensions into $l$ layers as by noting that

$$\boldsymbol{W}_{n \times m} = \boldsymbol{W}_{n \times p_{l-1}} \times \boldsymbol{W}_{p_{l-1} \times p_{l-2}} \times \cdots \times \boldsymbol{W}_{p_1 \times m} \,, \tag{4}$$

where we typically define $p_1 = rm$ and $p_i = rn, \forall i \neq 1$. In practice, considering the computational complexity of fully-connected layers, we advocate expanding each layer into only two or three layers with a small expansion rate. Note that this expansion is similar to that used in [5], which we discuss in

more detail in the supplementary material. However, as will be shown by our experiments, expanding *only* fully-connected layers, as in [5], does typically not yield a performance boost. By contrast, our two convolutional expansion strategies do.

Altogether, our strategies allow us to expand an arbitrary compact network into an equivalent deeper and wider one, and can be used independently or together. Once trained, the resulting ExpandNet can be contracted back to the original compact architecture in an algebraic manner, i.e., at no loss of information. Further implementation details are provided in the supplementary material.

## 4 Experiments

In this section, we demonstrate the benefits of our ExpandNets on image classification, object detection, and semantic segmentation. We further provide an ablation study to analyze the influence of different expansion strategies and expansion rates in the supplementary material.

We denote the expansion of convolutional layers by *CL*, of convolutional kernels by *CK*, and of fully-connected layers by *FC*. Specifically, we use *FC(Arora18)* to indicate that the expansion strategy is similar to the one used in [5]. When combining convolutional expansions with fully-connected ones, we use *CL+FC* or *CK+FC*.

### 4.1 Image Classification

We first study the use of our approach with very small networks on CIFAR-10 and CIFAR-100 [27], and then turn to the more challenging ImageNet [46] dataset, where we show that our method can improve the results of the compact MobileNet [21], MobileNetV2 [48] and ShuffleNetV2 $0.5\times$ [36].

#### 4.1.1 CIFAR-10 and CIFAR-100

**Experimental setup.** For CIFAR-10 and CIFAR-100 [27], we use the same compact network as in [37] (architecture and training setting in the supplementary material). To evaluate our kernel expansion method, we report results obtained with a similar network where the $3 \times 3$ kernels were replaced by $7 \times 7$ ones, with a padding of 3. In this set of experiments, the expansion rate $r$ is set to $4$ to balance the accuracy-efficiency trade-off. Since our expansion strategy is complementary to knowledge transfer, i.e., an ExpandNet can act as student in knowledge transfer, we further demonstrate its benefits in this setting by conducting experiments using knowledge distillation (KD) [20], hint-based transfer (Hint)[44] or probabilistic knowledge transfer (PKT) [37] from a ResNet18 teacher.

Table 1: Top-1 accuracy (%) of SmallNet with $7 \times 7$ kernels vs ExpandNets with $r = 4$ on CIFAR-10 and CIFAR-100.

| Model | Transfer | CIFAR-10 | CIFAR-100 |
|---|---|---|---|
| SmallNet | *w/o* KD | $78.63 \pm 0.41$ | $46.63 \pm 0.27$ |
| FC(Arora18) [5] | *w/o* KD | $78.64 \pm 0.39$ | $46.59 \pm 0.45$ |
| ACNet [11] | *w/o* KD | $79.37 \pm 0.52$ | $47.18 \pm 0.57$ |
| SmallNet | *w/* KD | $78.97 \pm 0.37$ | $47.04 \pm 0.35$ |
| ExpandNet-CL | | $78.47 \pm 0.20$ | $46.90 \pm 0.66$ |
| ExpandNet-CL+FC | *w/o* KD | $79.11 \pm 0.23$ | $46.66 \pm 0.43$ |
| ExpandNet-CK | | $80.27 \pm 0.24$ | $48.55 \pm 0.51$ |
| ExpandNet-CK+FC | | $\mathbf{80.31 \pm 0.27}$ | $\mathbf{48.62 \pm 0.47}$ |
| ExpandNet-CL+FC | *w/* KD | $79.60 \pm 0.25$ | $47.41 \pm 0.51$ |
| ExpandNet-CK+FC | | $\mathbf{80.63 \pm 0.31}$ | $\mathbf{49.13 \pm 0.45}$ |

We then evaluate our expansion strategies on MobileNet [21], MobileNetV2 [48], which we train for 350 epochs using a batch size of 128. We use stochastic gradient descent (SGD) with a momentum of 0.9, weight decay of 0.0005 and a learning rate of 0.1, divided by 10 at epochs 150 and 250. Note that training an ExpandNet takes slightly more time than training the compact network because of the extra parameters, as reported in Table 2. Therefore, to confirm that our better results are not just due to longer training, we also report the results of the baselines trained for the same amount of time as our ExpandNets.

**Results.** We first focus on the SmallNet with $7 \times 7$ kernels, for which we can evaluate all our expansion strategies, including the CK ones, and report the results of the model with $3 \times 3$ kernels in the supplementary material. Table 1 provides the results over 5 runs of all our networks with and without KD, which we have found to be the most effective knowledge transfer strategy, as evidenced by comparing

Table 2: Top-1 accuracy (%) of MobileNets vs ExpandNets with $r = 4$ on CIFAR-10 and CIFAR-100.

| Model | Epoch Time | CIFAR-10 | CIFAR-100 |
|---|---|---|---|
| MobileNet | 13.08s | 89.61 (88.87[†]) | 67.93 (68.18[†]) |
| ExpandNet-CL | 22.78s | **91.79** | **69.75** |
| MobileNetV2 | 24.88s | 91.64 (90.85[†]) | 71.66 (71.41[†]) |
| ExpandNet-CL | 49.22s | **92.58** | **72.33** |

† Accuracy with the same training time as ExpandNet-CL.
‡ Epoch Time was evaluated on CIFAR-10 on 2 32G TITAN V100 GPUs.

these results with those obtained by Hint and PKT in the supplementary material. As shown in the top portion of the table, only expanding the fully-connected layer, as in [5], yields mild improvement. However, expanding the convolutional ones clearly outperforms the compact network, and is further boosted by expanding the fully-connected one. Overall, expanding the kernels yields the best results; it outperforms even the concurrent convolutional expansion ACNet of [11]. Note that even without KD, our ExpandNets outperform SmallNet with KD. The gap is further increased when we also use KD, as shown in the bottom portion of the table.

In Table 2, we provide the results for MobileNet and MobileNetV2, including the baselines trained for a longer time, denoted by a †. These results confirm that our expansion strategies also boost the performance of these MobileNet models, even when the baselines are trained longer.

### 4.1.2 ImageNet

**Experimental setup**. For ImageNet [46], we use the compact MobileNet [21], MobileNetV2 [48] and ShuffleNetV2 [36] models, which were designed to be compact and yet achieve good results. We rely on a pytorch implementation of these models. For our approach, we use our CL strategy to expand all convolutional layers with kernel size $3 \times 3$ in

Table 3: Top-1 accuracy (%) on the ILSVRC2012 validation set (ExpandNets with $r = 4$).

| Model | MobileNet | MobileNetV2 | ShuffleNetV2 |
|---|---|---|---|
| original | 66.48 | 63.75 | 56.89 |
| ACNet [11] | 67.61 | 64.29 | 52.43 |
| original (*w/* KD) | 69.01 | 65.40 | 57.59 |
| ExpandNet-CL | **69.40** | **65.62** | **57.38** |
| ExpandNet-CL (*w/* KD) | **70.47** | **67.19** | **57.68** |

MobileNet and ShuffleNetV2, while only expanding the non-residual $3 \times 3$ convolutional layers in MobileNetV2. We trained the MobileNets using the short-term regime advocated in [17], i.e., 90 epochs with a weight decay of 0.0001 and an initial learning rate of 0.1, divided by 10 every 30 epochs. We employed SGD with a momentum of 0.9 and a batch size of 256. For ShuffleNet, we used the small ShuffleNetV2 $0.5\times$, trained as in [36]. We also applied KD from a ResNet152 (with 78.32% top-1 accuracy), tuning the KD hyper-parameters to the best accuracy for each method.

**Results.** We compare the results of the original models with those of our expanded versions in Table 3. Our expansion strategy increases the top-1 accuracy of MobileNet, MobileNetV2 and ShuffleNetV2 $0.5\times$ by 2.92, 1.87 and 0.49 percentage points (pp). It also yields consistently higher accuracy than the concurrent ACNet of [11]. Furthermore, our ExpandNets without KD outperform the MobileNets with KD, even though we do not require a teacher.

## 4.2 Object Detection

Our approach is not restricted to image classification. We demonstrate its benefits for one-stage object detection.

**Experimental setup.** YOLO-LITE [23], which was designed to work in constrained environments. The YOLO-LITE used here is very compact, consisting of a backbone with only 5 convolutional layers and of a head. We ex-

Table 4: YOLO-LITE vs ExpandNet with $r = 4$ on the PASCAL VOC2007 test set.

| Model | mAP (%) |
|---|---|
| YOLO-LITE | 27.34 |
| ExpandNet-CL | **30.97** |

panded the 5 backbone convolutional layers using our CL strategy with $r = 4$, and trained the resulting network on the PASCAL VOC2007 + 2012 [12, 13] training and validation sets in the standard YOLO fashion [39, 40]. We report the mean average precision (mAP) on the PASCAL VOC2007 test set.

**Results.** The results are reported in Table 4. As for object detection, our expansion strategy boosts the performance of the compact network. Specifically, we outperform it by over 3.5pp.

## 4.3 Semantic Segmentation

We then demonstrate the benefits of our approach on semantic segmentation using the Cityscapes dataset [8].

**Experimental setup.** For this experiment, we rely on the U-Net [45], which is a relatively compact network consisting of a contracting and an expansive path. We apply our CL expansion strategy with $r = 4$ to all convolutions

Table 5: U-Net vs ExpandNet with $r = 4$ on the Cityscapes validation set.

| Model | mIOU | mRec | mPrec |
|---|---|---|---|
| U-Net | 56.59 | 74.29 | 65.11 |
| ExpandNet-CL | **57.85** | **76.53** | **65.94** |

in the contracting path. We train the networks on 4 GPUs using the standard SGD optimizer with a

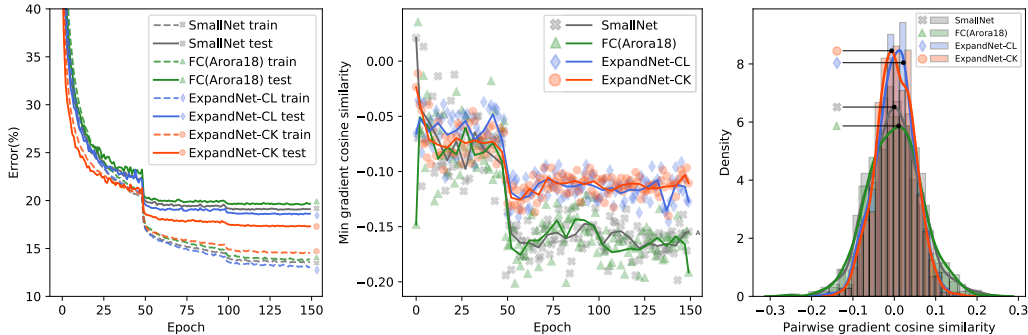

Figure 3: **Training behavior of networks with** $7 \times 7$ **kernels on CIFAR-10** (best viewed in color). *Left:* Training and test curves over 150 epochs. *Middle:* Minimum pairwise gradient cosine similarity at the end of each training epoch (higher is better). *Right:* Kernel density estimation of pairwise gradient cosine similarity at the end of training (over 5 independent runs).

Table 6: **Complexity analysis on CIFAR-10 for different expansion rates** $r$. The baseline network is the SmallNet with kernel size 7 (#Params:150.35K, #MACs: 6.12M, Epoch Time: 4.05s). Note that, for a given training setting, the wall-clock time only moderately increases as $r$ grows.

| $r$ | 2 | | | 4 | | | 8 | | |
|---|---|---|---|---|---|---|---|---|---|
| | #Params | #MACs | Epoch Time | #Params | #MACs | Epoch Time | #Params | #MACs | Epoch Time |
| FC(Arora18) | 339.40K | 6.30M | 4.09s | 675.91K | 6.64M | 3.94s | 1.74M | 7.70M | 4.02s |
| ExpandNet-CL | 562.95K | 25.16M | 4.13s | 2.17M | 98.39M | 4.61s | 8.58M | 389.35M | 9.39s |
| ExpandNet-CK | 237.72K | 14.38M | 4.10s | 653.25K | 42.64M | 4.12s | 2.07M | 141.10M | 5.50s |

momentum of 0.9 and a learning rate of $1e - 8$. Following the standard protocol, we report the mean Intersection over Union (mIOU), mean recall (mRec) and mean precision (mPrec).

**Results.** Our results on the Cityscapes validation set are shown in Table 5. Note that our ExpandNet outperforms the original compact U-Net.

## 5    Analysis of our Approach

To further analyze our approach, we first study its behavior during training and its generalization ability. For these experiments, we make use of the CIFAR-10 and CIFAR-100 datasets, and use the settings described in detail in our ablation study in the supplementary material. We then propose and analyze two hypotheses to empirically evidence that the better performance of our approach truly is a consequence of over-parameterization during training. In the supplementary material, we also showcase the use of our approach with the larger AlexNet architecture on ImageNet and evaluate the complexity of the models in terms of number of parameters, multiply-and-accumulate operations (MACs), and training and testing inference speed. Note that, since our ExpandNets can be contracted back to the original networks, *at test time, they have exactly the same number of parameters, MACs, and inference time as the original networks, but achieve better performance*.

### 5.1    Training Behavior

To investigate the benefits of linear over-parameterization on training, we make use of the gradient confusion introduced by Sankararaman et al. [49] to show that the gradients of nonlinearly over-parameterized networks were more consistent across mini-batches. Specifically, following [49], we measure gradient confusion (or rather consistency) as the minimum cosine similarity of gradients over 100 randomly-sampled pairs of mini-batches at the end of each training epoch. It measures the negative correlation between the gradients of different mini-batches, and thus indicates a disagreement on the parameter update. As in [49], we also combine the gradient cosine similarity of 100 pairs of sampled mini-batches at the end of training from each independent run and perform Gaussian kernel density estimation on this data.

We run each experiment 5 times and show the average values across all runs in Figure 3. The training and test curves show that our ExpandNets-CL/CK speed up convergence and yield a smaller generalization error. They also yield lower gradient confusion (higher minimum cosine similarity) and

Table 7: **Generalization ability on Corrupted CIFAR-10.** We report the top-1 error (%). Note that our ExpandNets yield smaller generalization errors than the compact network in almost all cases involving convolutional expansion. By contrast expanding FC layers often does not help.

| Dataset | Model | Kernel size $k$ | | | | | |
|---|---|---|---|---|---|---|---|
| | | 5 | | | 9 | | |
| | | Best Test | Last Test | Train | Best Test | Last Test | Train |
| 20% | SmallNet | $20.90 \pm 0.16$ | $21.09 \pm 0.20$ | $32.05 \pm 0.31$ | $22.56 \pm 0.39$ | $22.93 \pm 0.18$ | $29.61 \pm 0.36$ |
| | FC(Arora18) | $20.87 \pm 0.29$ | $21.06 \pm 0.26$ | $32.04 \pm 0.12$ | $22.95 \pm 0.39$ | $23.48 \pm 0.38$ | $29.83 \pm 0.34$ |
| | ExpandNet-CL | $20.47 \pm 0.46$ | $20.62 \pm 0.43$ | $31.80 \pm 0.23$ | $22.13 \pm 0.49$ | $22.73 \pm 0.53$ | $29.76 \pm 0.19$ |
| | ExpandNet-CK | $\mathbf{19.42 \pm 0.20}$ | $\mathbf{19.63 \pm 0.17}$ | $31.55 \pm 0.25$ | $\mathbf{19.32 \pm 0.31}$ | $\mathbf{19.55 \pm 0.30}$ | $\mathbf{31.65 \pm 0.17}$ |
| 50% | SmallNet | $25.38 \pm 0.45$ | $25.68 \pm 0.52$ | $54.49 \pm 0.41$ | $28.64 \pm 0.46$ | $30.44 \pm 0.57$ | $52.67 \pm 0.45$ |
| | FC(Arora18) | $25.36 \pm 0.63$ | $25.71 \pm 0.77$ | $54.44 \pm 0.08$ | $28.46 \pm 0.43$ | $30.89 \pm 0.38$ | $52.51 \pm 0.36$ |
| | ExpandNet-CL | $24.27 \pm 0.33$ | $24.63 \pm 0.44$ | $54.29 \pm 0.24$ | $27.42 \pm 0.35$ | $29.28 \pm 0.50$ | $52.67 \pm 0.27$ |
| | ExpandNet-CK | $\mathbf{22.82 \pm 0.27}$ | $\mathbf{23.00 \pm 0.29}$ | $53.93 \pm 0.23$ | $\mathbf{22.77 \pm 0.14}$ | $\mathbf{22.99 \pm 0.15}$ | $\mathbf{54.37 \pm 0.12}$ |
| 80% | SmallNet | $37.99 \pm 0.64$ | $39.33 \pm 0.75$ | $76.14 \pm 0.15$ | $41.73 \pm 0.58$ | $47.96 \pm 1.07$ | $74.01 \pm 0.32$ |
| | FC(Arora18) | $38.35 \pm 0.59$ | $39.61 \pm 0.87$ | $76.51 \pm 0.15$ | $42.31 \pm 0.46$ | $49.36 \pm 1.44$ | $74.59 \pm 0.35$ |
| | ExpandNet-CL | $36.75 \pm 0.64$ | $38.08 \pm 0.50$ | $76.09 \pm 0.11$ | $41.44 \pm 0.46$ | $46.75 \pm 0.49$ | $74.46 \pm 0.08$ |
| | ExpandNet-CK | $\mathbf{33.29 \pm 1.04}$ | $\mathbf{34.24 \pm 0.85}$ | $75.77 \pm 0.22$ | $\mathbf{33.29 \pm 0.58}$ | $\mathbf{33.75 \pm 0.49}$ | $\mathbf{76.27 \pm 0.23}$ |

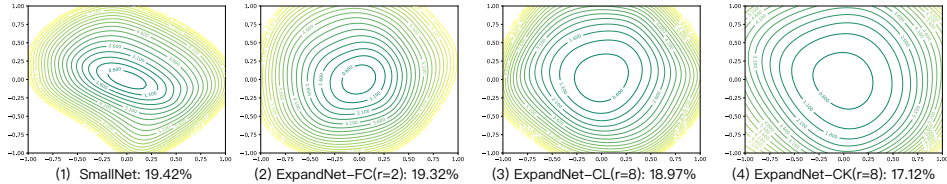

(1) SmallNet: 19.42%    (2) ExpandNet–FC(r=2): 19.32%    (3) ExpandNet–CL(r=8): 18.97%    (4) ExpandNet–CK(r=8): 17.12%

Figure 4: **Loss landscapes** of networks with $9 \times 9$ kernels on CIFAR-10 (We report top-1 error (%)).

a more zero-peaked density of pairwise gradient cosine similarity. This indicates that our networks are easier to train than the compact model. By contrast, only expanding the FC layers, as in [5], does not facilitate training. Additional plots are provided in the supplementary material.

**Computational overheads and complexity analysis.** To evaluate the influence of $r$ on the complexity of training, we report the number of parameters, MACs and wall-clock training time of a SmallNet with kernel size 7 on CIFAR-10 on a single 12G TITAN V. As shown in Table 6, our expansion strategies better leverage GPU computation, thus leading to only moderate wall-clock time increases as $r$ grows, particularly for our CK strategy. We provide further comparisons of the complexity of our expanded networks and of the original ones in terms of number of parameters, MACs and GPU speed with full use of GPUs in the supplementary material. Overall, as for very compact networks, our ExpandNets better exploit the GPU to make full use of its capacity, leading to similar training time to the original networks. For larger networks, such as MobileNets in Table 2, the GPU usage saturates, and thus the training time of ExpandNets increases. Nevertheless, since our ExpandNets can be contracted back to the original network, at test time, they have exactly the same number of parameters, MACs and inference time as the original networks, but our networks achieve better performance.

## 5.2 Generalization Ability

We then analyse the generalization ability of our approach. To this end, we first study the loss landscapes using the method in [33]. As shown in Figure 4, our ExpandNets with CL and CK expansion produce flatter minima, which, as discussed in [33], indicates better generalization.

As a second study of generalization, we evaluate the memorization ability of our ExpandNets on corrupted datasets, as suggested by Zhang et al. [62]. To this end, we utilize the open-source implementation of Zhang et al. [62] to generate three CIFAR-10 and CIFAR-100 training sets, containing 20%, 50% and 80% of random labels, respectively, while the test set remains clean.

In Table 7 (and Tables S7, S8 and S9 in the supplementary material), we report the top-1 test errors of the best model and of the one after the last epoch, as well as the training errors of the last model. These results evidence that CL and CK expansion typically yields lower test errors and higher training ones, which implies that our better results in the other experiments are not due to simply memorizing the datasets, but truly to better generalization ability.

## 5.3 Is Over-parameterization the Key to the Success?

In the previous experiments, we have shown the good training behavior and generalization ability of our expansion strategies. Below, we explore and reject two hypotheses other than over-parameterization that could be thought to explain our better results.

**Hypothesis 1**: The improvement comes from the different initialization resulting from expansion.

The standard (e.g., Kaiming) initialization of our ExpandNets is in fact equivalent to a non-standard initialization of the compact network. In other words, an alternative would consist of initializing the compact network with an un-trained algebraically-contracted ExpandNet. To investigate the influence of such different initialization schemes, we conduct several experiments on CIFAR-10, CIFAR-100 and ImageNet.

The results are provided in Table 8. On CIFAR-10, the compact networks initialized with FC(Arora18) and ExpandNet-CL yield slightly better results than training the corresponding ExpandNets. However, the same trend does not occur on CIFAR-100 and ImageNet, where, with ExpandNet initialization, MobileNet, MobileNetV2 and ShuffleNetV2

Table 8: Top-1 accuracy (%) of compact networks initialized with different ExpandNets on CIFAR-10, CIFAR-100 and ImageNet.

| Model | Initialization | CIFAR-10 | CIFAR-100 |
|---|---|---|---|
| SmallNet | Standard | $78.63 \pm 0.41$ | $46.63 \pm 0.27$ |
| | FC(Arora18) | $79.09 \pm 0.56$ | $46.52 \pm 0.36$ |
| | ExpandNet-CL | $78.65 \pm 0.36$ | $46.65 \pm 0.47$ |
| | ExpandNet-CL+FC | $78.81 \pm 0.52$ | $46.43 \pm 0.72$ |
| | ExpandNet-CK | $78.84 \pm 0.30$ | $46.56 \pm 0.23$ |
| | ExpandNet-CK+FC | $79.27 \pm 0.29$ | $46.62 \pm 0.29$ |
| ExpandNet-CK+FC | Standard | $\mathbf{80.31 \pm 0.27}$ | $\mathbf{48.62 \pm 0.47}$ |

| Model | Initialization | ImageNet |
|---|---|---|
| MobileNet | Standard | 66.48 |
| MobileNet | ExpandNet-CL | 66.44 |
| ExpandNet-CL | Standard | **69.40** |
| MobileNetV2 | Standard | 63.75 |
| MobileNetV2 | ExpandNet-CL | 63.07 |
| ExpandNet-CL | Standard | **65.62** |
| ShuffleNetV2 0.5× | Standard | 56.89 |
| ShuffleNetV2 0.5× | ExpandNet-CL | 56.91 |
| ExpandNet-CL | Standard | **57.38** |

$0.5\times$ reach results similar to or worse than standard initialization, while training ExpandNet-CL always outperforms the baselines. Moreover, the compact networks initialized by ExpandNet-CK always yield worse results than training ExpandNets-CK from scratch. This confirms that our results are not due to a non-standard initialization.

**Hypothesis 2**: The improvement is due to an intrinsic property of the CK expansion.

The amount of over-parameterization is directly related to the expansion rate $r$. Therefore, if some property of the CK strategy was the sole reason for our better results, and not over-parameterization, setting $r \leq 1$ should be sufficient. To study this, we follow the same experimental setting as for Table 1 but set $r \in \{0.25, 0.50, 0.75, 1.0, 2.0, 4.0\}$. As shown in Table 9, for $r < 1$, the performance of ExpandNet-CK drops by 6.38pp on CIFAR-10 and by 7.09pp on CIFAR-100 as the number of

Table 9: Top-1 accuracy (%) of SmallNet with $7 \times 7$ kernels vs ExpandNets with different $r$s on CIFAR-10 and CIFAR-100.

| $r$ | #params(K)$^\dagger$ | CIFAR-10 | CIFAR-100 |
|---|---|---|---|
| 0.25 | 37.91/43.76 | $72.32 \pm 0.62$ | $39.23 \pm 0.84$ |
| 0.50 | 42.81/48.66 | $76.77 \pm 0.36$ | $43.68 \pm 0.51$ |
| 0.75 | 48.43/54.28 | $78.70 \pm 0.42$ | $46.41 \pm 0.52$ |
| 1.00 | 54.77/60.62 | $79.22 \pm 0.52$ | $47.25 \pm 0.40$ |
| SmallNet | 66.19/72.04 | $78.63 \pm 0.41$ | $46.63 \pm 0.27$ |
| 2.0 | 87.32/93.17 | $79.97 \pm 0.18$ | $48.13 \pm 0.42$ |
| 4.0 | 187.0/192.8 | $\mathbf{80.27 \pm 0.24}$ | $\mathbf{48.55 \pm 0.51}$ |

$^\dagger$ #params(K) denotes the number of parameters (CIFAR-10 / CIFAR-100).

parameters decreases. For $r > 1$, ExpandNet-CK consistently outperforms SmallNet. Interestingly, with $r = 1$, ExpandNet-CK still yields better performance. This shows that our method benefits from both ExpandNet-CK and over-parameterization.

## 6 Conclusion

We have introduced an approach to training a given compact network from scratch by exploiting linear over-parameterization. Specifically, we have shown that over-parameterizing the network *linearly* facilitates the training of compact networks, particularly when linearly expanding convolutional layers. Our analysis has further evidenced that over-parameterization is the key to the success of our approach, improving both the training behavior and generalization ability of the networks, and ultimately leading to better performance at inference without any increase in computational cost. Our technique is general and can also be used in conjunction with knowledge transfer approaches to further boost performance. Finally, as shown in the supplementary material, initializing an ExpandNet with its trained nonlinear counterpart can further boost its results. This motivates us to investigate the design of other effective initialization schemes for compact networks in the future.

## Broader Impact

Our work introduces a general approach to improve the performance of a given compact convolutional neural network. It builds on the theoretical research on over-parameterization, but provides practical and effective ways to facilitate the training of convolutional layers, with extensive experiments and empirical analysis of our expansion strategies and of their impact on training behavior and generalization ability. Currently, our results on AlexNet in the supplementary material seem to indicate that expansion is not as effective on large networks than it is on compact ones. We nonetheless expect that our work will motivate other researchers to study solutions for this scenario.

Our approach is general, and thus applicable to a broad range of problems, including those demonstrated in our experiments, i.e., image classification, object detection and semantic segmentation, but not limited to them. In particular, because we focus on compact network, our work could have a significant impact for applications in resource-constrained environments, such as mobile phones, drones, or autonomous navigation. As a matter of fact, we are actively working on deploying our approach for perception-based autonomous driving. We acknowledge that such applications present security risks, e.g., related to adversarial attacks. We nonetheless expect these risks to be mitigated by the parallel research advances in adversarial robustness. Finally, from an ecological standpoint, our approach requires more training resources than the compact network, thus increasing its carbon footprint. Note, however, that this is mitigated by the fact that, at training time, we observed our expanded networks to make better use of the GPU resources than the compact ones.

## Acknowledgement

This work is supported in part by the Swiss National Science Foundation and by the Chinese Scholarship Council.

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
