[Supplementary Material]

# ExpandNets: Linear Over-parameterization to Train Compact Convolutional Networks – Supplementary Material –

## A Complementary Experiments

We provide additional experimental results to further evidence the effectiveness of our approach.

### A.1 Initializing ExpandNets

As demonstrated by our experiments in the main paper, training an ExpandNet from scratch yields consistently better results than training the original compact network. However, with deep networks, initialization can have an important effect on the final results. While designing an initialization strategy specifically for compact networks is an unexplored research direction, our ExpandNets can be initialized in a natural manner. To this end, we exploit the fact that an ExpandNet has a natural nonlinear counterpart, which can be obtained by incorporating a non-linear activation function between each pair of linear layers. We therefore propose to initialize the parameters of an ExpandNet by simply training its nonlinear counterpart and transferring the resulting parameters to the ExpandNet. The initialized ExpandNet is then trained in the standard manner.

Table S1: Top-1 accuracy (%) of SmallNet with $7 \times 7$ kernels vs ExpandNets with $r = 4$ on CIFAR-10 and CIFAR-100.

| Model | Transfer | CIFAR-10 | CIFAR-100 |
|---|---|---|---|
| SmallNet | *w/o* KD | $78.63 \pm 0.41$ | $46.63 \pm 0.27$ |
| FC(Arora18) [5] | *w/o* KD | $78.64 \pm 0.39$ | $46.59 \pm 0.45$ |
| ACNet [11] | *w/o* KD | $79.37 \pm 0.52$ | $47.18 \pm 0.57$ |
| SmallNet | *w/* KD | $78.97 \pm 0.37$ | $47.04 \pm 0.35$ |
| ExpandNet-CL | | $78.47 \pm 0.20$ | $46.90 \pm 0.66$ |
| ExpandNet-CL+FC | | $79.11 \pm 0.23$ | $46.66 \pm 0.43$ |
| ExpandNet-CL+FC+Init | *w/o* KD | $79.98 \pm 0.28$ | $47.98 \pm 0.48$ |
| ExpandNet-CK | | $80.27 \pm 0.24$ | $48.55 \pm 0.51$ |
| ExpandNet-CK+FC | | $\mathbf{80.31 \pm 0.27}$ | $\mathbf{48.62 \pm 0.47}$ |
| ExpandNet-CK+Init | | $\mathbf{80.81 \pm 0.27}$ | $\mathbf{49.82 \pm 0.25}$ |
| ExpandNet-CL+FC | | $79.60 \pm 0.25$ | $47.41 \pm 0.51$ |
| ExpandNet-CL+FC+Init | *w/* KD | $80.29 \pm 0.25$ | $48.62 \pm 0.34$ |
| ExpandNet-CK+FC | | $\mathbf{80.63 \pm 0.31}$ | $\mathbf{49.13 \pm 0.45}$ |
| ExpandNet-CK+FC+Init | | $\mathbf{81.21 \pm 0.17}$ | $\mathbf{50.37 \pm 0.39}$ |

We applied this initialization scheme to the SmallNets used in our CIFAR-10 and CIFAR-100 experiments, and report the results in Table S1 and S3, respectively, where +*Init* denotes the use of our initialization strategy. We also report the result of this initialization scheme on object detection in Table S2.

Table S2: YOLO-LITE vs ExpandNets with $r = 4$ on the PASCAL VOC2007 test set.

| Model | mAP(%) |
|---|---|
| YOLO-LITE | 27.34 |
| ExpandNet-CL | 30.97 |
| ExpandNet-CL+Init | **35.14** |

Note that this strategy yields an additional accuracy boost to our approach. In particular, since YOLO-LITE is very compact, this scheme boosts performance by more than 4pp.

Note that, on ImageNet and Cityscapes, the nonlinear counterparts of the ExpandNets did not outperform the ExpandNets, and thus we did not use our initialization strategy. As a general rule, when the nonlinear counterparts achieves better performance than the ExpandNets, we recommend using them for initialization. This suggests interesting directions for future research on the initialization of our ExpandNets and of compact networks in general.

### A.2 SmallNet with $3 \times 3$ Kernels on CIFAR-10 & CIFAR-100

As mentioned in the main paper, we also evaluate our approach using the same small network as in [37]. It is composed of 3 convolutional layers with $3 \times 3$ kernels and no padding. These 3 layers have 8, 16 and 32 output channels, respectively. Each of them is followed by a batch normalization layer, a ReLU layer and a $2 \times 2$ max pooling layer. The output of the last layer is passed through a fully-connected layer with 64 units, followed by a logit layer with either 10 or 100 units. All networks, including our ExpandNets, were trained for 150 epochs using

Table S3: Top-1 accuracy (%) of SmallNet with $3 \times 3$ kernels vs ExpandNets with $r = 4$ on CIFAR-10 and CIFAR-100.

| Model | Transfer | CIFAR-10 | CIFAR-100 |
|---|---|---|---|
| SmallNet | *w/o KD* | $73.32 \pm 0.20$ | $40.40 \pm 0.60$ |
| FC(Arora18) [5] | *w/o KD* | $73.78 \pm 0.83$ | $40.52 \pm 0.71$ |
| ACNet [11] | *w/o KD* | $74.52 \pm 0.30$ | $41.61 \pm 0.49$ |
| SmallNet | *w/ KD* | $73.34 \pm 0.31$ | $40.46 \pm 0.56$ |
| ExpandNet-CL | | $73.96 \pm 0.30$ | $40.91 \pm 0.47$ |
| ExpandNet-CL+FC | *w/o KD* | $74.45 \pm 0.29$ | $41.12 \pm 0.49$ |
| ExpandNet-CL+FC+Init | | $\mathbf{75.16 \pm 0.23}$ | $\mathbf{42.41 \pm 0.21}$ |
| ExpandNet-CL+FC | *w/ KD* | $74.52 \pm 0.37$ | $41.51 \pm 0.49$ |
| ExpandNet-CL+FC+Init | | $\mathbf{75.17 \pm 0.51}$ | $\mathbf{42.67 \pm 0.67}$ |

Table S5: **Small networks vs ExpandNets on CIFAR-10 (Top) and CIFAR-100 (Bottom).** We report the top-1 accuracy for the original compact networks and for different versions of our approach. Note that our ExpandNets yield higher accuracy than the compact network in almost all cases involving expanding convolutions. By contrast expanding FC layers does often not help.

| Model | $r$ | Kernel size $k$ | | | |
|---|---|---|---|---|---|
| | | 3 | 5 | 7 | 9 |
| SmallNet | | $79.34 \pm 0.42$ | $81.25 \pm 0.14$ | $81.44 \pm 0.20$ | $80.08 \pm 0.48$ |
| FC(Arora18) | 2 | $79.13 \pm 0.47$ | $81.26 \pm 0.33$ | $80.98 \pm 0.25$ | $80.43 \pm 0.22$ |
| | 4 | $78.92 \pm 0.36$ | $81.13 \pm 0.46$ | $80.85 \pm 0.24$ | $80.13 \pm 0.29$ |
| | 8 | $79.64 \pm 0.41$ | $81.21 \pm 0.18$ | $80.75 \pm 0.45$ | $80.16 \pm 0.16$ |
| ExpandNet-CL | 2 | $79.46 \pm 0.21$ | $81.50 \pm 0.31$ | $81.30 \pm 0.30$ | $80.26 \pm 0.66$ |
| | 4 | $\mathbf{79.90 \pm 0.21}$ | $81.60 \pm 0.15$ | $81.15 \pm 0.36$ | $80.62 \pm 0.32$ |
| | 8 | $79.78 \pm 0.20$ | $\mathbf{81.75 \pm 0.40}$ | $\mathbf{81.53 \pm 0.33}$ | $\mathbf{80.78 \pm 0.25}$ |
| ExpandNet-CK | 2 | $N/A$ | $81.72 \pm 0.31$ | $82.19 \pm 0.24$ | $81.60 \pm 0.11$ |
| | 4 | $N/A$ | $82.34 \pm 0.43$ | $82.34 \pm 0.22$ | $81.73 \pm 0.33$ |
| | 8 | $N/A$ | $\mathbf{82.37 \pm 0.25}$ | $\mathbf{82.84 \pm 0.28}$ | $\mathbf{82.53 \pm 0.30}$ |
| SmallNet | | $48.14 \pm 0.29$ | $50.44 \pm 0.07$ | $49.62 \pm 0.50$ | $48.70 \pm 0.38$ |
| FC(Arora18) | 2 | $47.21 \pm 0.46$ | $48.39 \pm 0.77$ | $47.88 \pm 0.41$ | $46.36 \pm 0.34$ |
| | 4 | $47.44 \pm 0.66$ | $48.92 \pm 0.47$ | $48.43 \pm 0.56$ | $46.90 \pm 0.34$ |
| | 8 | $47.55 \pm 0.25$ | $49.44 \pm 0.65$ | $48.66 \pm 0.49$ | $47.15 \pm 0.28$ |
| ExpandNet-CL | 2 | $47.68 \pm 0.85$ | $50.39 \pm 0.45$ | $49.78 \pm 0.33$ | $48.68 \pm 0.70$ |
| | 4 | $48.25 \pm 0.13$ | $50.68 \pm 0.27$ | $49.81 \pm 0.31$ | $\mathbf{48.87 \pm 0.65}$ |
| | 8 | $\mathbf{48.93 \pm 0.13}$ | $\mathbf{50.95 \pm 0.42}$ | $\mathbf{49.95 \pm 0.37}$ | $48.85 \pm 0.42$ |
| ExpandNet-CK | 2 | $N/A$ | $51.18 \pm 0.44$ | $51.09 \pm 0.41$ | $50.40 \pm 0.35$ |
| | 4 | $N/A$ | $\mathbf{52.13 \pm 0.36}$ | $51.82 \pm 0.67$ | $50.62 \pm 0.65$ |
| | 8 | $N/A$ | $52.05 \pm 0.59$ | $\mathbf{52.48 \pm 0.54}$ | $\mathbf{51.57 \pm 0.15}$ |

a batch size of 128. We used standard stochastic gradient descent (SGD) with a momentum of 0.9 and a learning rate of 0.01, divided by 10 at epochs 50 and 100. With this strategy, all networks reached convergence. For this set of experiments, we make use of our CL expansion strategy, with and without our initialization scheme discussed above, because the CK one does not apply to $3 \times 3$ kernels.

As reported in Table S3, expanding the convolutional layers yields higher accuracy than the small network. This is further improved by also expanding the fully-connected layer, and even more so when using our initialization strategy.

### A.3 Knowledge Transfer with ExpandNets

In the main paper, we claim that our ExpandNet strategy is complementary to knowledge transfer. Following [37], on CIFAR-10, we make use of the ResNet18 as teacher. Furthermore, we also use the same compact network with kernel size $3 \times 3$ and training setting as in [37]. In Table S4, we compare the results of different knowledge transfer strategies, including knowledge distillation (KD) [20], hint-based transfer (Hint)[44] and probabilistic knowledge transfer (PKT) [37], applied to the compact network and to our ExpandNets, without and with our initialization scheme. Note that using knowledge transfer with our ExpandNets, with and without initialization, consistently outperforms using it with the compact network. Altogether, we therefore believe that, to train a given compact network, one should really use both knowledge transfer and our ExpandNets to obtain the best results.

Table S4: **Knowledge transfer from the ResNet18 on CIFAR-10.** Using ExpandNets as student networks yields consistently better results than directly using SmallNet.

| Network | Transfer | Top-1 Accuracy |
|---|---|---|
| SmallNet | Baseline | $73.32 \pm 0.20$ |
| SmallNet | KD | $73.34 \pm 0.31$ |
| | Hint | $33.71 \pm 4.35$ |
| | PKT | $68.36 \pm 0.35$ |
| ExpandNet (CL+FC) | KD | $74.52 \pm 0.37$ |
| | Hint | $52.46 \pm 2.43$ |
| | PKT | $70.97 \pm 0.70$ |
| ExpandNet (CL+FC+Init) | KD | $\mathbf{75.17 \pm 0.51}$ |
| | Hint | $58.27 \pm 3.83$ |
| | PKT | $71.65 \pm 0.41$ |

### A.4 Hyper-parameter Choices

In this section, we evaluate the influence of the hyper-parameters of our approach, i.e., the expansion rate $r$ and the kernel size $k$. We study the behavior of our different expansion strategies, FC, CL and CK, separately, when varying the expansion rate $r \in \{2, 4, 8\}$ and the kernel size $k \in \{3, 5, 7, 9\}$. Compared to our previous CIFAR-10 and CIFAR-100 experiments, we use a deeper network with an

extra convolutional layer with 64 channels, followed by a batch normalization layer, a ReLU layer and a $2 \times 2$ max pooling layer. We use SGD with a momentum of 0.9 and a weight decay of 0.0005 for 150 epochs. The initial learning rate was 0.01, divided by 10 at epoch 50 and 100. Furthermore, we used zero-padding to keep the size of the input and output feature maps of each convolutional layer unchanged.

The results of these experiments are provided in Table S5. We observe that our different strategies to expand convolutional layers outperform the compact network in almost all cases, while only expanding fully-connected layers doesn't work well. In particular, for kernel sizes $k > 3$, ExpandNet-CK yields consistently higher accuracy than the corresponding compact network, independently of the expansion rate. For $k = 3$, where ExpandNet-CK is not applicable, ExpandNet-CL comes as an effective alternative, also consistently outperforming the baseline. In almost all cases, the performance of convolutional expansions improves as the expansion rate increases.

## A.5 Working with Larger Networks

We also evaluate the use of our approach with a larger network. To this end, we make use of AlexNet [28] on ImageNet. AlexNet relies on kernels of size 11 and 5 in its first two convolutional layers, which makes our CK expansion strategy applicable.

We use a modified, more compact version of AlexNet, where we replace the first fully-connected layer with a global average pooling layer, followed by a 1000-class fully-connected layer with softmax. To evaluate the impact of the network size, we explore the use of different dimensions, $[128, 256, 512]$, for the final convolutional features. We trained the resulting AlexNets and corresponding ExpandNets using the same training regime as for our MobileNets experiments in Section 4.

Table S6: **Top-1 accuracy** (%) **of AlexNet vs ExpandNets with** $r = 4$ **on the ILSVRC2012 validation set for different number of channels in the last convolutional layer.** Note that, while our expansion strategy always helps, its benefits decrease as the original model grows.

| # Channels | 128 | 256 (Original) | 512 |
|---|---|---|---|
| Baseline | 46.72 | 54.08 | 58.35 |
| ExpandNet-CK | **49.66** | **55.46** | **58.75** |
| ↑ | 2.94 | 1.38 | 0.4 |

As shown in Table S6, while our approach outperforms the baseline AlexNets for all feature dimensions, the benefits decrease as the feature dimension increases. This indicates that our approach is better suited for truly compact networks, and developing similar strategies for deeper ones will be the focus of our future research.

## A.6 Generalization Ability on Corrupted CIFAR-10 and CIFAR-100

Our experiments on corrupted datasets in the main paper imply better generalization. Here, we provide more experimental results on Corrupted CIFAR-10 (in Table S7) and CIFAR-100 (in Tables S8 and S9) by using different networks with kernel sizes of 3, 5, 7, 9, respectively. Our method consistently improves the generalization error gap across all kernel sizes and corruption rates (20%, 50%, 80%) and yields from around 1pp to over 6pp error drop in testing.

## A.7 Additional Visualizations

Here, we provide additional visualizations for the training behavior and the loss landscapes of Section 5, corresponding to networks with kernel sizes of 3, 5, 7, 9, respectively.

Table S7: Generalization ability (top-1 error (%)) on Corrupted CIFAR-10 (kernel size: 3, 7).

| Dataset | Model | Kernel size $k$ | | | | | |
|---|---|---|---|---|---|---|---|
| | | 3 | | | 7 | | |
| | | Best Test | Last Test | Train | Best Test | Last Test | Train |
| 20% | SmallNet | $22.20 \pm 0.43$ | $22.33 \pm 0.40$ | $34.85 \pm 0.18$ | $21.64 \pm 0.36$ | $21.98 \pm 0.42$ | $30.42 \pm 0.32$ |
| | FC(Arora18) | $22.40 \pm 0.29$ | $22.61 \pm 0.27$ | $35.12 \pm 0.07$ | $21.92 \pm 0.23$ | $22.35 \pm 0.43$ | $30.39 \pm 0.19$ |
| | ExpandNet-CL | $\mathbf{21.55 \pm 0.27}$ | $\mathbf{21.71 \pm 0.30}$ | $34.89 \pm 0.26$ | $21.25 \pm 0.41$ | $21.54 \pm 0.40$ | $30.36 \pm 0.24$ |
| | ExpandNet-CK | $N/A$ | $N/A$ | $N/A$ | $\mathbf{19.11 \pm 0.33}$ | $\mathbf{19.30 \pm 0.35}$ | $\mathbf{31.14 \pm 0.11}$ |
| 50% | SmallNet | $25.74 \pm 0.25$ | $25.94 \pm 0.15$ | $56.48 \pm 0.25$ | $26.99 \pm 0.69$ | $27.87 \pm 0.71$ | $53.27 \pm 0.21$ |
| | FC(Arora18) | $25.54 \pm 0.47$ | $25.80 \pm 0.41$ | $56.37 \pm 0.15$ | $26.86 \pm 0.46$ | $28.23 \pm 0.61$ | $53.14 \pm 0.20$ |
| | ExpandNet-CL | $\mathbf{25.48 \pm 0.35}$ | $\mathbf{25.66 \pm 0.43}$ | $56.41 \pm 0.33$ | $26.05 \pm 0.31$ | $26.99 \pm 0.15$ | $53.21 \pm 0.16$ |
| | ExpandNet-CK | $N/A$ | $N/A$ | $N/A$ | $\mathbf{22.43 \pm 0.47}$ | $\mathbf{22.61 \pm 0.49}$ | $\mathbf{53.74 \pm 0.16}$ |
| 80% | SmallNet | $37.49 \pm 0.62$ | $37.87 \pm 0.63$ | $77.46 \pm 0.16$ | $39.08 \pm 0.41$ | $43.33 \pm 0.77$ | $74.69 \pm 0.26$ |
| | FC(Arora18) | $37.26 \pm 0.16$ | $37.63 \pm 0.14$ | $77.54 \pm 0.07$ | $40.51 \pm 0.39$ | $44.82 \pm 0.62$ | $75.38 \pm 0.23$ |
| | ExpandNet-CL | $\mathbf{35.86 \pm 0.43}$ | $\mathbf{36.05 \pm 0.44}$ | $77.56 \pm 0.11$ | $39.40 \pm 0.93$ | $42.77 \pm 0.96$ | $75.24 \pm 0.22$ |
| | ExpandNet-CK | $N/A$ | $N/A$ | $N/A$ | $\mathbf{32.62 \pm 0.28}$ | $\mathbf{33.86 \pm 0.37}$ | $\mathbf{75.65 \pm 0.16}$ |

Table S8: Generalization ability (top-1 error (%)) on Corrupted CIFAR-100 (kernel size: 3, 5).

| Dataset | Model | Kernel size $k$ | | | | | |
| --- | --- | --- | --- | --- | --- | --- | --- |
| | | 3 | | | 5 | | |
| | | Best Test | Last Test | Train | Best Test | Last Test | Train |
| 20% | SmallNet | $55.30 \pm 0.42$ | $55.48 \pm 0.41$ | $62.20 \pm 0.31$ | $53.95 \pm 0.33$ | $54.16 \pm 0.34$ | $58.53 \pm 0.30$ |
| | FC(Arora18) | $56.15 \pm 0.22$ | $56.35 \pm 0.23$ | $62.60 \pm 0.19$ | $54.84 \pm 0.71$ | $55.05 \pm 0.76$ | $59.47 \pm 0.32$ |
| | ExpandNet-CL | $\mathbf{54.85 \pm 0.27}$ | $\mathbf{55.04 \pm 0.33}$ | $61.62 \pm 0.43$ | $53.50 \pm 0.35$ | $53.71 \pm 0.38$ | $58.09 \pm 0.31$ |
| | ExpandNet-CK | $N/A$ | $N/A$ | $N/A$ | $\mathbf{51.98 \pm 0.28}$ | $\mathbf{52.10 \pm 0.26}$ | $57.67 \pm 0.63$ |
| 50% | SmallNet | $62.54 \pm 0.74$ | $62.71 \pm 0.75$ | $78.81 \pm 0.39$ | $61.84 \pm 0.29$ | $62.16 \pm 0.21$ | $76.78 \pm 0.28$ |
| | FC(Arora18) | $63.65 \pm 0.50$ | $63.88 \pm 0.47$ | $79.40 \pm 0.18$ | $62.99 \pm 0.69$ | $63.21 \pm 0.60$ | $77.85 \pm 0.31$ |
| | ExpandNet-CL | $\mathbf{61.95 \pm 0.61}$ | $\mathbf{62.11 \pm 0.59}$ | $78.78 \pm 0.48$ | $61.49 \pm 0.39$ | $61.70 \pm 0.43$ | $76.73 \pm 0.26$ |
| | ExpandNet-CK | $N/A$ | $N/A$ | $N/A$ | $\mathbf{58.96 \pm 0.32}$ | $\mathbf{59.14 \pm 0.41}$ | $76.24 \pm 0.30$ |
| 80% | SmallNet | $78.35 \pm 0.83$ | $78.52 \pm 0.86$ | $93.78 \pm 0.18$ | $78.59 \pm 0.27$ | $78.81 \pm 0.35$ | $93.10 \pm 0.12$ |
| | FC(Arora18) | $80.36 \pm 0.55$ | $80.47 \pm 0.55$ | $94.38 \pm 0.12$ | $80.97 \pm 0.51$ | $81.15 \pm 0.53$ | $94.10 \pm 0.16$ |
| | ExpandNet-CL | $79.44 \pm 0.72$ | $79.66 \pm 0.75$ | $94.02 \pm 0.16$ | $79.87 \pm 0.29$ | $80.04 \pm 0.29$ | $93.59 \pm 0.20$ |
| | ExpandNet-CK | $N/A$ | $N/A$ | $N/A$ | $\mathbf{77.22 \pm 0.47}$ | $\mathbf{77.38 \pm 0.41}$ | $93.15 \pm 0.25$ |

Table S9: Generalization ability (top-1 error (%)) on Corrupted CIFAR-100 (kernel size: 7, 9).

| Dataset | Model | Kernel size $k$ | | | | | |
| --- | --- | --- | --- | --- | --- | --- | --- |
| | | 7 | | | 9 | | |
| | | Best Test | Last Test | Train | Best Test | Last Test | Train |
| 20% | SmallNet | $55.36 \pm 0.44$ | $55.66 \pm 0.43$ | $56.33 \pm 0.51$ | $56.59 \pm 0.72$ | $57.26 \pm 0.64$ | $55.16 \pm 0.32$ |
| | FC(Arora18) | $56.31 \pm 0.78$ | $56.58 \pm 0.77$ | $57.93 \pm 0.29$ | $57.82 \pm 0.23$ | $58.07 \pm 0.22$ | $57.09 \pm 0.52$ |
| | ExpandNet-CL | $54.87 \pm 0.47$ | $55.22 \pm 0.55$ | $55.52 \pm 0.49$ | $56.05 \pm 0.75$ | $56.51 \pm 0.76$ | $54.99 \pm 0.48$ |
| | ExpandNet-CK | $\mathbf{51.24 \pm 0.60}$ | $\mathbf{51.40 \pm 0.66}$ | $\mathbf{56.40 \pm 0.21}$ | $\mathbf{52.36 \pm 0.54}$ | $\mathbf{52.55 \pm 0.47}$ | $57.76 \pm 0.50$ |
| 50% | SmallNet | $63.76 \pm 0.59$ | $64.08 \pm 0.58$ | $75.45 \pm 0.23$ | $64.83 \pm 0.41$ | $65.63 \pm 0.40$ | $75.21 \pm 0.31$ |
| | FC(Arora18) | $64.54 \pm 0.72$ | $64.91 \pm 0.55$ | $76.75 \pm 0.39$ | $66.11 \pm 0.45$ | $66.73 \pm 0.41$ | $76.44 \pm 0.40$ |
| | ExpandNet-CL | $63.36 \pm 0.49$ | $63.73 \pm 0.54$ | $75.25 \pm 0.45$ | $64.36 \pm 0.54$ | $65.25 \pm 0.37$ | $74.84 \pm 0.29$ |
| | ExpandNet-CK | $\mathbf{58.74 \pm 0.25}$ | $\mathbf{58.98 \pm 0.20}$ | $75.24 \pm 0.24$ | $\mathbf{60.42 \pm 0.86}$ | $\mathbf{60.65 \pm 0.86}$ | $76.73 \pm 0.40$ |
| 80% | SmallNet | $79.73 \pm 0.47$ | $79.95 \pm 0.36$ | $92.58 \pm 0.20$ | $81.02 \pm 0.92$ | $81.70 \pm 0.97$ | $92.54 \pm 0.26$ |
| | FC(Arora18) | $82.97 \pm 0.83$ | $83.20 \pm 0.83$ | $94.13 \pm 0.34$ | $83.42 \pm 0.72$ | $83.82 \pm 0.71$ | $93.94 \pm 0.35$ |
| | ExpandNet-CL | $80.79 \pm 0.54$ | $81.09 \pm 0.62$ | $93.22 \pm 0.30$ | $81.02 \pm 0.44$ | $81.58 \pm 0.46$ | $93.25 \pm 0.56$ |
| | ExpandNet-CK | $\mathbf{78.51 \pm 0.41}$ | $\mathbf{78.64 \pm 0.36}$ | $93.24 \pm 0.15$ | $\mathbf{80.15 \pm 0.50}$ | $\mathbf{80.32 \pm 0.55}$ | $94.04 \pm 0.21$ |

Figure S1: Loss landscape plots on CIFAR-10 (We report top-1 error (%)).

We plot the loss landscapes of SmallNets and corresponding ExpandNets on CIFAR-10 in Figure S1, and analyze the training behavior on CIFAR-10 in Figure S2 and on CIFAR-100 in Figure S3. These plots further confirm that in almost all cases, our convolution expansion strategies (*CL* and *CK*) facilitate training (with lower gradient confusion and more 0-concentrated gradient cosine similarity density) and yield better generalization ability (with flatter minima).

(a) kernel size: 3

(b) kernel size: 5

(c) kernel size: 7

(d) kernel size: 9

Figure S2: **Training behavior of networks on CIFAR-10** (best viewed in color). *Left:* Training and test curves over 150 epochs. *Middle:* Minimum pairwise gradient cosine similarity at the end of each training epoch (higher is better). *Right:* Kernel density estimation of pairwise gradient cosine similarity at the end of training (over 5 independent runs).

(a) kernel size: 3

(b) kernel size: 5

(c) kernel size: 7

(d) kernel size: 9

Figure S3: **Training behavior of networks on CIFAR-100** (best viewed in color). *Left:* Training and test curves over 150 epochs. *Middle:* Minimum pairwise gradient cosine similarity at the end of each training epoch (higher is better). *Right:* Kernel density estimation of pairwise gradient cosine similarity at the end of training (over 5 independent runs).

Table S10: **ExpandNet complexity analysis on CIFAR-10, ImageNet, PASCAL VOC and Cityscapes.** Note that, within each task, the metrics are the same for all networks, since we can compress our ExpandNets back to the small network.

| Model | # Params(M) | | # MACs | | GPU Speed (imgs/sec) | |
|---|---|---|---|---|---|---|
| | Train | Test | Train | Test | Train | Test |
| SmallNet ($7 \times 7$) | 0.07 | | 4.49M | | 147822.51 | |
| ExpandNet-CL | 0.55 | | 57.49M | | 64651.81 | |
| ExpandNet-CL+FC | 2.11 | 0.07 | 59.04M | 4.49M | 61379.95 | 154850.52 |
| ExpandNet-CK | 0.19 | | 23.95M | | 75065.09 | |
| ExpandNet-CK+FC | 1.75 | | 25.5M | | 68679.89 | |
| MobileNet | 4.23 | 4.23 | 0.58G | 0.58G | 3797.21 | 3829.81 |
| ExpandNet-CL | 4.96 | | 1.76G | | 729.78 | |
| MobileNetV2 | 3.50 | 3.50 | 0.32G | 0.32G | 3417.20 | 3419.43 |
| ExpandNet-CL | 3.67 | | 1.34G | | 1009.25 | |
| ShuffleNetV2 $0.5\times$ | 1.37 | 1.37 | 0.04G | 0.04G | 5404.06 | 5434.58 |
| ExpandNet-CL | 1.41 | | 0.6G | | 4228.10 | |
| YOLO-LITE | 0.57 | 0.57 | 1.81G | 1.81G | 7.94 | 19.82 |
| ExpandNet-CL | 4.48 | | 28.59G | | 6.07 | |
| U-Net | 7.76 | 7.76 | 389.26G | 389.26G | 8.25 | 8.25 |
| ExpandNet-CL | 82.97 | | 2586.02G | | 2.98 | |

# B   Complexity Analysis

Here, we compare the complexity of our expanded networks and the original networks in terms of number of parameters, number of MACs and GPU speed.

In Table S10, we provide numbers for both training and testing. During training, because our approach introduces more parameters, inference is 2 to 5 times slower than in the original network for an expansion rate of 4. However, since our ExpandNets can be contracted back to the original network, at test time, they have exactly the same number of parameters and MACs, and inference time, but our networks achieve higher accuracy.

# C   Discussion of Related Methods

In this section, we discuss in more detail the two works that are most closely related to ours. These two works also evidence the benefits of linear over-parameterization, thus strengthening our argument, but differ significantly from ours in terms of specific strategy. Note that, as shown by our experiments, our approach outperforms theirs.

## C.1   Discussion of [5]

Arora et al. [5] worked mostly with purely linear, fully-connected models, with only one example using a nonlinear model, where again only the fully-connected layer was expanded. By contrast, we focus on *practical, nonlinear, compact convolutional* networks, and we propose two ways to expand convolutional layers, which have not been studied before. As shown by our experiments in the main paper and in Section A of this supplementary material, our convolutional linear expansion strategies yield better solutions than vanilla training, with higher accuracy, more zero-centered gradient cosine similarity during training and minima that generalize better. This is in general not the case when expanding the fully-connected layers only, as proposed by Arora et al. [5]. Furthermore, in contrast with [5], who only argue that depth speeds up convergence, we empirically show, by using different expansion rates, that increasing width helps to reach better solutions. We now discuss in more detail the only experiment in [5] with a nonlinear network.

In their paper, Arora et al. performed a sanity test on MNIST with a CNN, but only expanding the fully-connected layer. According to our experiments, expanding fully-connected layers only (denoted as FC(Arora18) in our results) is typically insufficient to outperform vanilla training of the compact network. This was confirmed by using their code, with which we found that, in their setting, the over-parameterized model yields higher test error. We acknowledge that Arora et al. [5] did not claim that expansion led to better results but sped up convergence. Nevertheless, this seemed to contradict our experiments, in which our FC expansion was achieving better results than that of Arora et al. [5].

Figure S4: **Product $L^2$ vs Normal $L^2$** (best viewed in color). *Left:* Training curves of the overall loss function. *Middle Left:* Training curves of the cross-entropy. *Middle Right:* Curves of training errors. *Right:* Curves of test errors. (Note that the $y$-axis is in log scale.)

While analyzing the reasons for this, we found that Arora et al. [5] used a different weight decay regularizer than us. Specifically, considering a single fully-connected layer expanded into two, this regularizer is defined as

$$L_r = \|\tilde{\boldsymbol{W}}_{fc}\|_2^2 = \|\boldsymbol{W}_{fc_1}\boldsymbol{W}_{fc_2}\|_2^2 \, , \qquad (S.1)$$

where $\boldsymbol{W}_{fc_1}$ and $\boldsymbol{W}_{fc_2}$ represent the parameter matrices of the two fully-connected layers after expansion. That is, the regularizer is defined over the *product* of these parameter matrices. While this corresponds to weight decay on the original parameter matrix, without expansion, it contrasts with usual weight decay, which sums over the different parameter matrices, yielding a regularizer of the form

$$L_r = \|\boldsymbol{W}_{fc_1}\|_2^2 + \|\boldsymbol{W}_{fc_2}\|_2^2 \, . \qquad (S.2)$$

The product $L^2$ norm regularizer used by Arora et al. [5] imposes weaker constraints on the individual parameter matrices, and we observed their over-parameterized model to converge to a worse minimum and lead to worse test performance when used in a nonlinear CNN.

To evidence this, in Figure S4, we compare the original model with an over-parameterized one relying on a product $L^2$ regularizer as in [5], and with an over-parameterized network with normal $L^2$ regularization, corresponding to our FC expansion strategy. Even though the overall loss of Arora et al. [5]'s over-parameterized model decreases faster than that of the baseline, the cross-entropy loss term, the training error and the test error do not show the same trend. The test errors of the original model, Arora et al. [5]'s over-parameterized model with product $L^2$ norm and our ExpandNet-FC with normal $L^2$ norm are 0.9%, 1.1% and 0.8%, respectively. Furthermore, we also compare Arora et al. [5]'s over-parameterized model and our ExpandNet-FC with an expansion rate $r = 2$. We observe that Arora et al. [5]'s over-parameterized model performs even worse with a larger expansion rate, while our ExpandNet-FC works well.

Note that, in the experiments in the main paper and below, the models denoted by FC(Arora18) rely on a normal $L^2$ regularizer, which we observed to yield better results and makes the comparison fair as all models then use the same regularization strategy.

## C.2  Discussion of ACNet [11]

The work of Ding et al. [11], concurrent to ours, also proposed a form of expansion of convolutions. Specifically, as shown in Figure S5, their approach consists of replacing a convolutional layer with $k \times k$ kernels with three parallel layers: One with the same square $k \times k$ kernel, and two with 1D asymmetric convolutions of size $1 \times k$ and $k \times 1$. These three different convolutions are then applied in parallel on the same input feature map, and their outputs are combined via addition.

Figure S5: **One ACNet block** (best viewed in color).

As argued in [11], the goal of this operation is to increase the representation power of a standard square kernel by strengthening the kernel skeletons. While effective, the over-parameterization resulting from this approach remains limited; by using 1D convolutions in parallel to the original ones, it can only add $2kmn$ parameters for every $k \times k$ kernel with $m$ input and $n$ output channels. By contrast, by incorporating new convolutional layers in a serial manner, we can modify the number of channels of the intermediate layers so as to increase the number of parameters of the network much more

drastically, and in a much more flexible way, thanks to our expansion rate. Specifically, with an expansion rate $r$, our CL expansion strategy yield $rm^2 + k^2r^2mn + rn^2$ parameters instead of $k^2mn$ for the original convolution. Ultimately, while ACNet can indeed improve the image classification performance, as shown in [11] and confirmed by our experiments, the greater flexibility of our approach yields significantly better results, particularly for networks relying on depthwise convolutions, as evidenced by our ImageNet results, and networks with kernel sizes larger than 3, as evidenced by our results with a SmallNet with $7 \times 7$ kernels. Furthermore, note that, in contrast to Arora et al. [5] and Ding et al. [11], we also demonstrate the effectiveness of our expansion strategy on object detection and semantic segmentation.

## D    Matrix Representation of a Convolution Operator

We provide an example of the matrix representation of a convolutional layer, following Eq. 1 in the main paper. Given an input $\mathbf{X}_{1 \times 1 \times 3 \times 3}$ and convolutional filters $\mathbf{F}_{1 \times 1 \times 2 \times 2}$, expressed as

$$\mathbf{X}_{1 \times 1 \times 3 \times 3} = \left[ \left[ \begin{bmatrix} x_{11} & x_{12} & x_{13} \\ x_{21} & x_{22} & x_{23} \\ x_{31} & x_{32} & x_{33} \end{bmatrix} \right] \right] , \quad \mathbf{F}_{1 \times 1 \times 2 \times 2} = \left[ \left[ \begin{bmatrix} k_{11} & k_{12} \\ k_{21} & k_{22} \end{bmatrix} \right] \right] , \tag{S.3}$$

the matrix representation of a convolution can be obtained by vectorizing the input as

$$\boldsymbol{X}^v_{9 \times 1} = \begin{bmatrix} x_{11} & x_{12} & x_{13} & x_{21} & x_{22} & x_{23} & x_{31} & x_{32} & x_{33} \end{bmatrix}^T , \tag{S.4}$$

and by defining a highly-structured matrix containing the filters as

$$\boldsymbol{W}^{\mathsf{F}}_{4 \times 9} = \begin{bmatrix} k_{11} & k_{12} & 0 & k_{21} & k_{22} & 0 & 0 & 0 & 0 \\ 0 & k_{11} & k_{12} & 0 & k_{21} & k_{22} & 0 & 0 & 0 \\ 0 & 0 & 0 & k_{11} & k_{12} & 0 & k_{21} & k_{22} & 0 \\ 0 & 0 & 0 & 0 & k_{11} & k_{12} & 0 & k_{21} & k_{22} \end{bmatrix} . \tag{S.5}$$

Then, the convolution operation $(*)$ can be equivalently written as

$$\mathbf{Y}_{1 \times 1 \times 2 \times 2} = \mathbf{X}_{1 \times 1 \times 3 \times 3} * \mathbf{F}_{1 \times 1 \times 2 \times 2} = \mathrm{reshape}(\boldsymbol{W}^{\mathsf{F}}_{4 \times 9} \times \boldsymbol{X}^v_{9 \times 1}) , \tag{S.6}$$

where $\times$ denotes the standard matrix-vector product.

To contract an ExpandNet, one can then compute the matrix product of its expanded layers to obtain a single matrix representing these multiple operations. This matrix can then be transferred back to a standard convolution filter tensor representation following the reverse strategy to the one explained above. For the details of how we contract our ExpandNets in practice, we invite the reader to check our submitted code (codesource/exp_cifar/utils/compute_new_weights.py and codesource/exp_imagenet/utils/compute_new_weights.py). Note that, in our implementation, we take advantage of the Pytorch tensor operators.

Below, we provide some toy code to expand a convolutional layer with either standard or depthwise convolutions and contract the expanded layers back. This code is based on our submitted code and can also be found in codesource/dummy_test.py.

Pytorch code to expand and contract back a standard convolutional layer:

```
import torch
import torch.nn as nn

m = 3   # input channels
n = 8   # output channels
k = 5   # kernel size
r = int(4)   # expansion rate
imgs = torch.randn((8, m, 7, 7))   # input images with batch size as 8

# original standard convolutional layer
F = nn.Conv2d(m, n, k)

# Expand-CL with r
F1 = nn.Conv2d(m, r*m, 1)
```

```
15  F2 = nn.Conv2d(r*m, r*n, k)
16  F3 = nn.Conv2d(r*n, n, 1)
17
18  # contracting
19  from exp_cifar.utils.compute_new_weights \
20      import compute_cl, compute_cl_2
21  tmp = compute_cl(F1, F2)
22  tmp = compute_cl_2(tmp, F3)
23  F.weight.data, F.bias.data = tmp['weight'], tmp['bias']
24
25  # test
26  res_cl = F3(F2(F1(imgs)))
27  res_F = F(imgs)
28  print('Contract from Expand-CL: %.7f' % (res_cl-res_F).sum())# <10^-5
29
30  # Expand-CK
31  # k = 5, l=2
32  F1 = nn.Conv2d(m, r*m, 3)
33  F2 = nn.Conv2d(r*m, n, 3)
34
35  # contracting
36  from exp_cifar.utils.compute_new_weights import compute_ck
37  tmp = compute_ck(F1, F2)
38  F.weight.data = tmp['weight']
39  F.bias.data = tmp['bias']
40
41  # test
42  res_ck = F2(F1(imgs))
43  res_F = F(imgs)
44  print('Contract from Expand-CK: %.7f' % (res_ck-res_F).sum())# <10^-5
```
Listing 1: Expansion and contraction of a standard convolutional layer

Pytorch code to expand and contract back a depthwise convolutional layer with a kernel size of 3:

```
1  # for depthwise conv, input channels=out channels
2  m = 4   # input channels
3  n = 4   # output channels
4  k = 3   # kernel size
5  r = int(4)   # expansion rate
6  imgs = torch.randn((8, m, 7, 7))
7
8  # original depthwise convolutional layer
9  F = nn.Conv2d(m, n, k, groups=m, bias=False)
10
11 # Expand-CL with r
12 F1 = nn.Conv2d(m, r*m, 1, groups=m, bias=False)
13 F2 = nn.Conv2d(r*m, r*m, k, groups=m, bias=False)
14 F3 = nn.Conv2d(r*m, n, 1, groups=m, bias=False)
15
16 # contracting
17 from exp_imagenet.utils.compute_new_weights \
18     import compute_cl_dw_group, compute_cl_dw_group_2
19
20 tmp = compute_cl_dw_group(F1, F2)
21 tmp = compute_cl_dw_group_2(tmp, F3)
22
23 F.weight.data = tmp['weight']
24
25 # test
26 res_depthwise_cl = F3(F2(F1(imgs)))
27 res_F = F(imgs)
28 print('Contract from depthwise Expand-CL: %.7f' % (res_depthwise_cl-
        res_F).sum())# <10^-5
```
Listing 2: Expansion and contraction of a depthwise convolutional layer