[Reviews · NeurIPS 2020]

Review 1

Summary and Contributions: Post-rebuttal: Thank you for the rebuttal. I have maintained a recommendation for accept but lowered my score from 8 to 7 after reading the other reviewers' critiques. Pre-rebuttal: The authors conduct an empirical study into the effects of linear overparameterisation in nonlinear CNNs. They find that overparameterisation can lead to modest performance gains. They also conduct ablation studies into how overparameterisation affects generalisation error, the gradient distribution, and minima sharpness/flatness. They also run some experiments targetted at pinpointing the exact cause of performance improvements and conclude it is indeed overparametisation.

Strengths: - The central message of the paper is clear: they explore overparameterisation and find it helps. - The methods are straightforward and I can see such simple methods as easily adopted by the community. I think this sort of work is relevant in its applicability. - The experiments appears quite thorough and I believe I could reimplement them. - My favourite part of the paper is the ablation studies, where I felt I learnt the most. A couple of key phenomena are observed, such as tighter generalisation gaps, improved gradient confusion, and reduced side-lobes in pairwise gradient similarity. - I also really enjoyed page 8 concerning the two rejected hypotheses. This sort of empirical work, in my eyes, is very valuable for the community and adds significance to the paper.

Weaknesses: - I do not have any great criticisms of this paper. On the whole I am very happy with it. My main criticisms are on two experiments (gradient analysis and initialisation), which can be read in the next section. - I guess the paper lacks in novelty, but that not important since it is compensated by good empirical analysis

Correctness: The mathematics of the main method seems sound. The empirical methodology all seems sound to me as well, apart from two parts 1) In the analysis of the minimum gradient confusion and the cosine gradient similarity, which gradients were analysed exactly? While the results are interesting they may not be meaningful if either a) not taken from equivalent points in the networks, or b) not descriptive of the training of the all the layers in each network as a whole. 2) For initialisation it is shown that contracting a network from an ExpandNet leads to an equivalently initialised SmallNet. Indeed this is true for the forward pass of the network, but are the gradients in the backward pass exactly the same? Initialisation is typically designed to limit the explosion/vanishing of both forward activations and backwards gradients, but the authors have just considered the forward pass. Maybe it is true for the backward pass too, I don't know. Knowing this would help distinguish whether it is the gradients themselves which differ between an equivalent ExpandNet and SmallNet or whether it is the training dynamics (i.e. the optimizer, which we know not to be covariant to gradient reparameterisation).

Clarity: The paper is well-written and easy to understand. The authors clearly define the scope and context of the work against the backdrop of contemporary literature.

Relation to Prior Work: The paper is clearly depicted again the backdrop of past and concurrent works and in the experimental section pointers are made to other works of note.

Reproducibility: Yes

Additional Feedback: Is there a definition of "compact"? Perhaps because I have my mathematics hat on I keep on thinking it may mean something technical that I don't know, but I'm guessing the authors just mean contracted/compressed post-overparameterisation? In Equation 1 is there are bijection between the original representation of a convolutional kernel and the "matrixised" form? I think so, but it would be nice to know that this is true. I would like to see a short discussion on the computational overhead/complexity of training in terms of expansion rate r. Line 147: what is i in the equation p = r^i m and why would this lead to an explosion in parameters? Surely if r and/or i is close to unity it should not be much of an issue? Line 157: It may be useful to note that k is odd or instead of l = (k-1)/2 just write k = 2l+1. Padding and strides: is the correspondence between the padding/striding schemes and the contracted versions exact? A single line stating this, if true, would be useful. What is the logic behind including knowledge distillation in the experiments? I can see that this serves to improve performance, but I also think that this somewhat detracts from (what I see is) the core message of the paper, which is to explore the benefits of overparameterisation. Tables with standard deviations: Good job for including standard deviations, but does it make sense to bolden the results with the highest mean, when the standard deviations show overlap with the next few highest models? I would suggest to bolden the multiple highest performing models, which have significant overlap or to remove the boldening. Did the authors record wall-clock training times anywhere? I would like to see these. Line 262: Does 3.5pp means 3.5 %? Cityscapes dataset: the mIOU reported are quite a lot lower than the current models reports (~ 84%). Could this be due to very small learning rate of 1e-8? Figure 4: Are the CK and CL minima really that much flatter than for SmallNet? Since this is a qualitative result I would be cautious in offering this as an explanation for why overparameterisation leads to better overall performance. Line 353: SmalleNet -> SmallNet


Review 2

Summary and Contributions: This paper proposes an expanding strategy (ExpandNets) to facilitate the training of compact convolutional networks. Both the fully-connected and convolutional layers in networks are expanded to deeper ones with more parameters during training, but equivalently converted to the original layers for inference. Positive results are achieved with ExpandNets in image classification, semantic segmentation and object detection. The authors also empirically shown that ExpandNets accelerate training.

Strengths: It is interesting to leverage the benefits of over-parameterization during training but improves the efficiency for inference with less parameters. The empirical results on three tasks seem strong. ExpandNets are shown to produce flatter minima and reduce the variants of gradients. Besides, the effectiveness of over-parameterization is studied, which may motivate further research.

Weaknesses: Since the proposed method increases the channels of each convolutional layer by r times (ignoring the two 1x1 conv layers), ExpandNets may suffer from a high computation complexity during training. I am concerned about the time and memory cost. In addition, it may be unfair to compare the results with baselines that actually use less training time. For example, network pruning also train large models but inference with small models. There are not such comparisons with these methods. The reported performance of MobileNetV2 on ImageNet is much weaker than the original paper [48] (63.75% v.s. 72.00 %). The baseline models seem to be rather weak. For example, the accuracy on CIFAR-10 is about 80%, while the current SOTA has reached >98%. I’m not saying that SOTA accuracy should be pursued, but the presented results are cleared not competitive and not convincing.

Correctness: The claims and the empirical methodology are correct.

Clarity: This paper is well written, and easy to follow.

Relation to Prior Work: The differences of this work from previous ones are clearly discussed.

Reproducibility: Yes

Additional Feedback:


Review 3

Summary and Contributions: The author proposed a novel method to train a compact CNN, referred to as ExpandNets. The proposed method is simple but effective. Furthermore, the proposed method has the advantage that it can be combined with existing training methods such as knowledge distillation. The author conducted various experiments to validate the effectiveness of the proposed method. Experiments included image classification, object detection, image segmentation, and many ablation studies.

Strengths: - The paper is well organized and easy to follow. - The most novel point of the proposed method is the expanding scheme for the convolutional layers. The proposed scheme amplifies the number of parameters by increasing the number of channels via 1 by 1 convolution or replacing convolutional layers with consecutive 3 by 3 convolutional layers. These techniques have been generally used to change the number of parameters. However, the idea of expansion of linear layer for training and contraction for testing to improve the performance seems to be new. - The experimental results of submitted paper show that just expanding the linear layers (conv., fc) can improve performance of the model. I think this is meaningful results. The author empirically verified the effectiveness of the proposed model.

Weaknesses: - I wonder why the proposed method works well. The paper claims that the proposed method works well because of over-parameterization through parameter expansion. This was experimentally verified through self-ablation (section 5.1). However, there are still questions. In order for networks of the same structure to exhibit different performance, the network parameter (weight) must be learned differently. I would like to see why more beneficial the back-propagation of the expanded layers are than the non-expanded layer. It would be required to mathematically analyze how the parameter update of the linear layer changes depending on the expansion of linear layer. - Generally, it is considered that the power of deep neural networks comes from a non-linearity of the model. However, after contracting the expanded linear layers (conv., fc) into one layer, the network is equivalent to the network with the non-expended layered network. This implies there is a way to get the learning results via the expanded layers. I suspect that the backpropagation settings for the non-expanded network has not been established to the best ones. If not, it should be needed why the use of expanded layers (over-parameterization) is beneficial to the improvement. it would be clear to give a mathematical explanation for this. - The idea of simplifying the multiple linear layer into a single layer has been proposed in `Fekix Wu, et al., Simplifying Graph Convolution Networks, ICML 2019.

Correctness: There are no technical errors.

Clarity: The paper is well organized and easy to follow. However, the key claim is not supported by rationale as mentioned in Weakness.

Relation to Prior Work: The idea of simplifying the multiple linear layer into a single layer has been proposed in `Fekix Wu, et al., Simplifying Graph Convolution Networks, ICML 2019. Even though it is applied to graph neural network, the idea is the same.

Reproducibility: Yes

Additional Feedback: The authors should provide rationale on why the use of expanded layers (over-parameterization) is beneficial to the promotion of training of non-expanded network contrary to the direct training of it.


Review 4

Summary and Contributions: This paper proposes a training strategy to train a given compact network. Given an arbitrary compact network, the paper expands the network into an over-parameterized one. Three expansion methods are introduced, i.e., expanding convolutional layers (CL), convolutional kernels (CK) and fc layers (FC). Experiments show that using the proposed network over-parameterization strategies, a small network can be better trained than training the original network.

Strengths: The proposed expansion methods are technically reasonable and easy to understand. The expansion techniques are based on existing network factorization methods, but the authors use them to improve the training of a given fixed network which differs from the previous ones trying to compress networks with matrix factorization. Such an over-parameterization scheme seems to be easy to be applied to any compact networks. The experimental results on image classification, object detection and semantic segmentation broadly validates the efficacy of the methods.

Weaknesses: I think the expansion techniques are not of strong novelty, as indicated above. The expansion of CL and FC are quite common.

Correctness: The methods look technically sound.

Clarity: Yes.

Relation to Prior Work: Yes.

Reproducibility: Yes

Additional Feedback:

[Author Response · NeurIPS 2020]

Table 1: Complexity analysis on CIFAR-10 for different expansion rates $r$. (The original network is the SmallNet with kernel size 7 used in Section 5 (#Params:150.35K, #MACs: 6.12M, Epoch Time: 4.05s).

| $r$ | 2 | | | 4 | | | 8 | | |
|---|---|---|---|---|---|---|---|---|---|
| | #Params | #MACs | Epoch Time | #Params | #MACs | Epoch Time | #Params | #MACs | Epoch Time |
| FC(Arora18) | 339.40K | 6.30M | 4.09s | 675.91K | 6.64M | 3.94s | 1.74M | 7.70M | 4.02s |
| ExpandNet-CL | 562.95K | 25.16M | 4.13s | 2.17M | 98.39M | 4.61s | 8.58M | 389.35M | 9.39s |
| ExpandNet-CK | 237.72K | 14.38M | 4.10s | 653.25K | 42.64M | 4.12s | 2.07M | 141.10M | 5.50s |

**To R1:** Thank you for the positive feedback. We will incorporate your suggestions in the final version.

**Gradient analysis.** Following [49], we account for all the layers in each network to analyze the gradient confusion.

**Initialization.** The gradients of the initialized ExpandNet before and after contraction are not the same, which indeed
contributes to their different training behavior. This was studied in [5] for purely linear FC networks, and remains true
for our new convolutional expansion strategies in non-linear CNNs. We will clarify this.

**Computational overheads and wall-clock training times.** The numbers provided in Table 1 above for a SmallNet
with kernel size 7 show that our expansion strategies can better leverage GPU computation, thus leading to only
moderate wall-clock time increases as $r$ grows, particularly for our CK strategy. We report additional values for
MobileNets in Table 2 below and provide a complexity analysis in Section B of the supplementary material.

**Other suggestions:** As a rule of thumb, "compact networks" in practice have < 10% of the parameters of conventional
CNNs. There is indeed a bijection between the original and matrix form of a convolution. The padding/stride schemes
maintain equivalence between the expanded and compact networks.

**To R2: Training time.** As shown in Table 1 above,
the training time only moderately increases with $r$, be-
cause our models can better exploit the GPU to make
full use of its capacity, particularly for very small net-
works. For larger ones, the GPU usage saturates, and
thus the training time of ExpandNets increases. To
nonetheless confirm that our better results are not just
due to longer training, we increased the training time
of the MobileNets to match that of our ExpandNets-CL.

Table 2: Top-1 accuracy (%) of MobileNets vs ExpandNets with $r = 4$ on CIFAR-10 and CIFAR-100.

| Model | #Params | #MACs | Epoch Time | CIFAR-10 | CIFAR-100 |
|---|---|---|---|---|---|
| MobileNet | 3.22M | 47.18M | 13.08s | 89.61 (88.87†) | 67.93 (68.18†) |
| ExpandNet-CL | 3.94M | 101.36M | 22.78s | **91.79** | **69.75** |
| MobileNetV2 | 2.30M | 94.60M | 24.88s | 91.64 (90.85†) | 71.66 (71.41†) |
| ExpandNet-CL | 3.12M | 207.87M | 49.22s | **92.58** | **72.33** |

† Accuracy with the same training time as ExpandNet-CL.
‡ #Params, #MACs and Epoch Time were evaluated on CIFAR-10 on 2 32G TITAN V100 GPUs.

This led to the accuracies highlighted in Table 2 with a †, which remain lower than those of our ExpandNets.

**Network pruning.** Our work tackles a different problem from network pruning: 1) We aim to train a given compact
architecture, whereas pruning yields an arbitrary one, which precludes a direct comparison; 2) We show that linear
over-parameterization is beneficial for network training; 3) Contraction in our work yields no information loss.

**MobileNetV2 on ImageNet.** Many people have observed that training MobileNetV2 on ImageNet following the
experimental setting in [48] does not yield the reported results. In a recent repository[1], heavy hyper-parameter tuning
eventually led to 72.20% accuracy. In this setting, our ExpandNet-CL reaches *73.16%*.

**CIFAR baselines.** We focused on compact networks, which may not yield SOTA results, such as the SmallNet ($\sim$80%
on CIFAR-10) taken from [37]. Nonetheless, in Table 2, we report the results of *new*, stronger baselines – MobileNets
on the CIFAR datasets. These results, which we will include in the final version, show that stronger baselines still
benefit from our expansion strategies.

**To R3: Rationale of our method.** Over-parameterization is widely acknowledged as beneficial for network training,
and mathematical explanations have been studied in [1, 2, 5, 16, 50, 62] for linear networks under specific assumptions.
Here, we contribute novel linear expansion strategies for convolutional networks and confirm the benefits of over-
parameterization. Note that, as mentioned in Answer 2 to R1, our expansion schemes modify the network gradients,
thus explaining their different training behavior. Importantly, all other reviewers acknowledge the thoroughness and
interest of our empirical validation, and argue that our work is significant for the community.

**Backpropagation settings of non-expanded networks.** As mentioned to R2, we used the hyper-parameters tuned for
the non-expanded networks to train both the non-expanded networks and our ExpandNets. Despite this, our ExpandNets
still reach better solutions. This remains true for our new MobileNetV2 experiments on ImageNet and CIFAR above.

**Felix Wu et al..** Thank you for the reference, which we will discuss. Wu et al. show that non-linearities can be removed
from the *MLP* layers of graph convolution networks. Because they collapse multiple FC layers into one, their work is in
fact closer to [5]. In contrast to our work, they do *not* remove the non-linearities of the conv layers and collapse them;
do *not* introduce expansion strategies for conv layers; and do *not* study the benefits of linear over-parameterization.

**To R4:** Thank you for the positive feedback. As correctly pointed out by R4, our work differs from matrix factorization
in that, via our expansion strategies, we aim to incorporate over-parameterization in a given compact network so as to
improve its accuracy. We will highlight this in the final version.

## Footnotes

[1] Reference: d-li14, PyTorch Implemention of MobileNetV2, available at https://github.com/d-li14/mobilenetv2.pytorch.


[Meta-Review · NeurIPS 2020]

The paper shows that if a certain layer of a NN is expanded as a composition of multiple linear layers, it leads to higher performance than training the reduced network (i.e., when multiple layers are reduced to a single layer). The results are convincing and the ablation analysis is thorough. I encourage the authors to further probe if the same results hold true for standard architectures like ResNet and if there are theoretical insights to be gleaned from the empirical findings.